# Boosted Catalytic Performance of Ni$_2$Co$_2$@T-PMo@ZIF-67 for Glucose Oxidation in a Direct-Glucose Fuel Cell

Shipu Jiao [1,†], Ning Kang [1,†], Miao Liu [1], Yihao Zhang [1], Yang Li [1], Bushra Maryam [1], Xu Zhang [1], Pingping Zhang [2,*] and Xianhua Liu [1,*]

[1] School of Environmental Science and Engineering, Tianjin University, Tianjin 300354, China; jsp@tju.edu.cn (S.J.); 2120214001@tju.edu.cn (N.K.); liumiao@tju.edu.cn (M.L.); zhangyihao_@tju.edu.cn (Y.Z.); liyang_@tju.edu.cn (Y.L.); maryambushra@yahoo.com (B.M.); zhangxu_2022@tju.edu.cn (X.Z.)

[2] College of Food Science and Engineering, Tianjin Agricultural University, Tianjin 300384, China

[*] Correspondence: zpp@tjau.edu.cn (P.Z.); lxh@tju.edu.cn (X.L.)

[†] These authors contributed equally to this work.

**Abstract:** In this study, we report on how to design efficient catalysts for glucose oxidation via the transitional metal doping of nanohybrids of polyoxometalates (POMs) and metal-organic frameworks (MOFs). ZIF-67, a cobalt-based MOF, as well as phosphomolybdic acid (PMo), were used as precursors for the fabrication of pyrolyzed PMo@ZIF-67 (T-PMo@ZIF-67). A different amount of Ni$^{2+}$ was doped into PMo@ZIF-67 to produce Ni$_x$Co$_y$@T-PMo@ZIF-67. Among them, Ni$_2$Co$_2$@T-PMo@ZIF-67 had the best performance. The power density of the fuel cell that used Ni$_2$Co$_2$@T-PMo@ZIF-67 as an anode catalyst was 3.76 times that of the cell that used active carbon as an anode catalyst. SEM and EDS mapping results indicate that Ni$_2$Co$_2$@T-PMo@ZIF-67 has a spherical structure and rough surface, and elements such as cobalt, nickel, and molybdenum are evenly distributed. XRD characterization indicates that Co$_3$O$_4$, CoMoO$_4$, CoNiO$_4$, and MoNiO$_4$ co-exist in the composites. It is supposed that Co$^{2+}$, Mo$^{6+}$, and Ni$^{2+}$ in the composites may have synergistic effects on the catalytic oxidation of glucose.

**Keywords:** MOF derivatives; polyoxometalates; fuel cells; catalysis





## 1. Introduction

Metal-organic frameworks (MOFs) are a class of crystalline porous materials with a periodic network structure formed by interconnecting inorganic metal centers (metal ions or metal clusters) with bridging organic ligands through self-assembly. In MOFs, the organic ligands, as well as metal ions or clusters, are arranged with an obvious orientation, which can form different framework pore structures, thus exhibiting different adsorption properties, optical properties, electromagnetic properties, catalytic properties, etc. [1,2]. The preparation of MOF-derived materials performed via pyrolysis using metal-organic framework materials (MOFs) as templates, or precursors, has become a popular method for catalyst development [3–6]. The integration of MOFs into various materials, such as metal nanoparticles and semiconductors, can further improve the performance of MOF derivatives, but the selection of specific metal precursors and MOFs ligands is very limited, so the obtained functional materials do not meet the increasing needs of diverse applications [7]. Polyoxometalate (POM) is a negatively charged transition metal ion aggregate formed in a high oxygen state, and has a special and superior three-dimensional structure, as well as physical and chemical properties [8–10]. The structure, surface chemistry, polarity, charge, and redox properties of POMs can be easily tailored by the introduction of organic moieties and metal-organic groups. However, their electrochemical applications are limited due to their moderate electrochemical performance and low specific surface area. MOFs have the advantages of high porosity, large specific surface area, good chemical stability, and multiple adsorption sites. They can enclose POMs in their metal-organic skeleton so

as to avoid the loss of POMs and prolong the catalyst service life [11,12]. Therefore, the combination of MOFs and POMs is expected to expand the availability of transition metal elements, as well as improve the performance of MOF-derived materials [13–17].

With the gradual depletion of fossil resources, and the aggravated environmental deterioration, the development of novel technologies for sustainable energy conversion and storage has attracted global attention [18–21]. Glucose fuel cells (GFC) are one type of fuel cells that produce electricity by oxidizing glucose at the anode and reducing oxygen at the cathode. Glucose is the most abundant monosaccharide in nature, and is cheap and easy to obtain, non-volatile, and safe. It is also environmentally friendly, green, and non-toxic [22]. Glucose fuel cells can be divided into two categories according to the operating environment: external fuel cells and implantable fuel cells. In the case of external operation, glucose fuel cells can directly use agricultural waste as an energy source for power generation, which has the advantages of simplicity, safety, environmental protection, and wide fuel sources. Implantable fuel cells use glucose in body fluids, such as interstitial fluid, tears, blood, and cerebrospinal fluid to power the corresponding implanted devices. They have broad application prospects in implantable medical devices such as cardiac pacemakers and drug delivery pumps [23,24]. Generally, glucose is harder to oxidize than conventional substrates of fuel cells such as hydrogen and methanol, and thus needs an efficient and selective catalyst [25,26]. The anodic glucose oxidation reaction is slow, and the substrate is difficult to react completely [27,28]. The catalyst can improve the anodic oxidation rate of glucose fuel cells [29–32]. Currently, Pt is considered to be the most efficient catalyst for various fuel cells, but high Pt loading are required to meet practical applications [33,34]. As a result, fuel cells are too expensive to be commercialized on a large scale. It is a great challenge to develop high performance and low-cost anode catalysts for GFCs [35].

In this paper, we used phosphomolybdic acid (PMo, a keggin-type POM), as well as ZIF-67 (a cobalt-based MOF), as precursors for the fabrication of pyrolyzed PMo@ZIF-67 (T-PMo@ZIF-67). We tried to enhance their catalytic performance by doping various transitional metal ions ($Ni^{2+}$, $Fe^{3+}$, $Cu^{2+}$, and $Zn^{2+}$) in the preparation process of T-PMo@ZIF-67. The effects of doping with different transitional metal ions on the electrochemical performance were investigated. In addition, the possible mechanism of glucose oxidation under an alkaline environment was also discussed.

## 2. Results and Discussion

### 2.1. Electrochemical Performance Characterization of Anodic Catalysts

2.1.1. Linear Sweep Voltammetry (LSV) Measurement

Figure 1a shows the LSV plots of the different T-XPMo@ZIF-67 anodes, where the current density of T-NiPMo@ZIF-67 was always higher than that of T-PMo@ZIF-67. The current density of T-CuPMo@ZIF-67 and T-FePMo@ZIF-67 were lower than that of T-PMo@ZIF-67 at the same potential, and the current density of T-ZnPMo@ZIF-67 was even lower than the blank control anode (AC) at the potential range of −0.46 V~0.3 V. The oxidation current increased in the order of T-ZnPMo@ZIF-67 < AC < T-FePMo@ZIF-67 < T-CuPMo@ZIF-67 < T-PMo@ZIF-67 < T-NiPMo@ZIF-67. This indicates that $Zn^{2+}$ remarkably impairs the catalytic activity of T-PMo@ZIF-67. It can be inferred that $Zn^{2+}$ has a lower catalytic activity towards glucose oxidation. The addition of $Zn^{2+}$ may precede the formation of ZIF-8 material by $Co^{2+}$ during the preparation process, resulting in the loss of $Co^{2+}$ during the preparation process and the reduction of $Co^{2+}$ involved in the reaction. Although $Cu^{2+}$ and $Fe^{2+}$ can promote the glucose oxidation reaction, the catalytic performances of T-CuPMo@ZIF-67 and T-FePMo@ZIF-67 were not as high as that of T-PMo@ZIF-67. The doping of $Ni^{2+}$ in the catalyst significantly increased the oxidation current of the glucose.

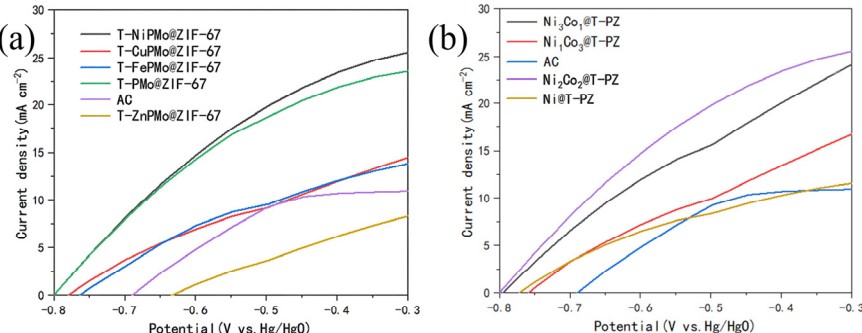

**Figure 1.** (**a**) LSV diagram of T-XPMo@ZIF-67 doped anodes; (**b**) LSV diagram of $Ni_xCo_y$@ZIF-67 doped anodes (10 mL 3 M of KOH and 1 M of glucose solution).

In order to optimize the catalytic performance of T-NiPMo@ZIF-67, the $Co^{2+}/Ni^{2+}$ ratio was tailored in the synthesis process of $Ni_xCo_y$@T-PZ. Figure 1b shows the LSV measurements of the catalysts with different $Co^{2+}/Ni^{2+}$ ratios. The current densities of the four anodes were higher than that of the AC control, and the maximum current density was increased in the order of AC < Ni@T-PZ < Ni1Co3@T-PZ < $Ni_3Co_1$@T-PZ < $Ni_2Co_2$@T-PZ. This result further confirmed that doping with appropriate amounts of $Ni^{2+}$ (1.25 mM) can promote the oxidation of glucose, and the optimal $Co^{2+}/Ni^{2+}$ ratio as a molar ratio was 1:1. $Co^{2+}$ and $Ni^{2+}$ may have a synergistic effect on the catalytic oxidation capacity of glucose. Therefore, the LSV curves of $Ni_3Co_1$@T-PZ and $Ni_2Co_2$@T-PZ were higher than that of Ni@T-PZ, which Ni2+ was added alone. However, when the concentration of either metal ion was higher, the catalytic performance of the catalyst would deteriorate.

### 2.1.2. Electrochemical Impedance Testing (EIS) of Anodes

EIS tests were performed on anodes modified with different catalysts. The resulting data were fitted by the equivalent circuit diagram (Figure S1). Figure 2a shows the EIS curves corresponding to blank anodes (AC), T-PMo@ZIF-67, T-NiPMo@ZIF-67, T-CuPMo@ZIF-67, T-FePMo@ZIF-67, and T-ZnPMo@ZIF-67 catalyst doped anodes. Table 1 shows the impedance values of each part of the anode modified by different anode catalysts. As can be seen from the figure, the performance of T-NiPMo@ZIF-67 was significantly better than that of blank AC anode and other metal ion catalysts, showing the smallest Ohmic resistance and charge transfer resistance. The difference between the several electrodes was that the metal ions added were different, which indicates that the type of metal ions has an important effect on the conductivity of the electrode surface.

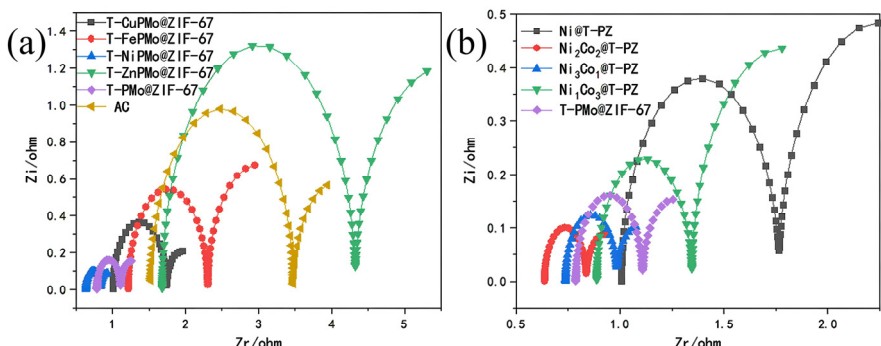

**Figure 2.** (**a**) EIS diagram of the different T-XPMo@ZIF-67 anodes; (**b**) EIS diagram of the $Ni_xCo_y$@T-PZ anodes (10 mL 3 M of KOH and 1 M of glucose solution).

In order to better study the effects of $Co^{2+}/Ni^{2+}$ ratio on the performance of T-NiPMo@ZIF-67, the impedance of the anodes with different $Co^{2+}/Ni^{2+}$ ratios were analyzed. Figure 2b and Table 2 show the Rct was increased in the order of $Ni_2Co_2$@T-PZ < $Ni_3Co_1$@T-PZ < T-PMo@ZIF-67 < $Ni_1Co_3$@T-PZ < Ni@T-PZ. As can be seen from Table 2,

changes of Rs of the anodes with different $Co^{2+}/Ni^{2+}$ ratios were consistent with Rct. $Ni_2Co_2$@T-PZ had the smallest Rs of 0.686 Ω. The Rct of $Ni_2Co_2$@T-PZ (0.1755 Ω) was decreased by 42.70% when compared with T-PMo@ZIF-67 (0.3063 Ω), and decreased by 76.85% when compared with Ni@T-PZ (0.7582 Ω). Compared with T-PMo@ZIF-67 and Ni@T-PZ, the total resistance of $Ni_2Co_2$@T-PZ decreased by 24.88% and 61.10%, respectively. These data can explain why the performance of $Ni_2Co_2$@T-PZ is better than that of other catalysts. The decrease of transfer resistance accelerated the electron transfer and improved the current flow.

**Table 1.** Impedance values of T-XPMo@ZIF-67 anodes.

|  | AC | T-CuPMo@ZIF-67 | T-FePMo@ZIF-67 | T-NiPMo@ZIF-67 | T-ZnPMo@ZIF-67 | T-PMo@ZIF-67 |
|---|---|---|---|---|---|---|
| Rs (Ω) | 1.518 | 1.002 | 1.162 | 0.686 | 1.68 | 0.7869 |
| Rct (Ω) | 1.151 | 0.4188 | 0.6594 | 0.1755 | 2.643 | 0.3063 |
| Rd (Ω) | 1.957 | 0.7333 | 0.7416 | 0.201 | 2.403 | 0.3212 |
| Rt (Ω) | 4.626 | 2.1541 | 2.563 | 1.0625 | 6.726 | 1.4144 |

**Table 2.** Impedance values of $Ni_xCo_y$@T-PZ anodes.

|  | Ni@T-PZ | $Ni_1Co_3$@T-PZ | $Ni_3Co_1$@T-PZ | $Ni_2Co_2$@T-PZ | T-PMo@ZIF-67 |
|---|---|---|---|---|---|
| Rs (Ω) | 1.006 | 0.887 | 0.738 | 0.686 | 0.7869 |
| Rct (Ω) | 0.7582 | 0.4577 | 0.202 | 0.1755 | 0.3063 |
| Rd (Ω) | 0.967 | 0.8711 | 0.2467 | 0.201 | 0.3212 |
| Rt (Ω) | 2.7312 | 2.2158 | 1.1867 | 1.0625 | 1.4144 |

### 2.1.3. The Tafel Curve of Anodes

The Tafel curve is a section of the strong polarization region of the general polarization curve. For the simpler electron transfer process, the Tafel curve can be used for analysis, and the exchange current density can be obtained through using the linear part of the Tafel curve to intersect the axis through the extension line, so as to calculate the number of electrons transferred in the electrochemical process. Figure 3a shows the Tafel curve of different catalyst modified anodes, and Figure 3b shows the fitting curves. Table 3 shows the fitting results of the Tafel measurement. As can be seen from Figure 3b and Table 3, The Tafel slopes of AC, T-PMo@ZIF-67, Ni@T-PZ, $Ni_3Co_1$@T-PZ, $Ni_2Co_2$@T-PZ, and $Ni_1Co_3$@T-PZ were 186.1702, 207.3794, 184.5195, 197.7402, 194.4443 and 191.6377 mV dec$^{-1}$ respectively. The $I_0$ of the different anodes were increased in the order of AC ($3.831 \times 10^{-4}$ A cm$^{-2}$) < Ni@T-PZ ($8.283 \times 10^{-4}$ A cm$^{-2}$) < $Ni_1Co_3$@T-PZ ($10.117 \times 10^{-4}$ A cm$^{-2}$) < T-PMo@ZIF-67 ($24.526 \times 10^{-4}$ A cm$^{-2}$) < $Ni_3Co_1$@T-PZ ($34.901 \times 10^{-4}$ A cm$^{-2}$) < $Ni_2Co_2$@T-PZ ($37.884 \times 10^{-4}$ A cm$^{-2}$). $Ni_2Co_2$@T-PZ had the highest exchange current density, which was much higher than that of the AC anode. This indicates that the glucose oxidation reaction of $Ni_2Co_2$@T-PZ catalyst doped anode can be carried out at a lower overpotential. It was confirmed that $Ni_2Co_2$@T-PZ can significantly promote the oxidation of glucose, increase the reaction rate, and improve the performance of fuel cell.

**Table 3.** Tafel curve fitting results.

|  | Fitting Equation | $R^2$ | Tafel (mV dec$^{-1}$) | $10^{-4}I_0$ (A cm$^{-2}$) |
|---|---|---|---|---|
| AC | y = 5.37143 × −3.41673 | 0.9983 | 186.1702 | 3.831 |
| Ni@T-PZ | y = 5.41948 × −3.08181 | 0.99818 | 184.5195 | 8.283 |
| $Ni_3Co_1$@T-PZ | y = 5.05714 × −2.45716 | 0.99876 | 197.7402 | 34.901 |
| $Ni_2Co_2$@T-PZ | y = 5.14286 × −2.42154 | 0.99758 | 194.4443 | 37.884 |
| $Ni_1Co_3$@T-PZ | y = 5.21818 × −2.99496 | 0.99807 | 191.6377 | 10.117 |
| T-PMo@ZIF-67 | y = 4.82208 × −2.61037 | 0.99766 | 207.3794 | 24.526 |

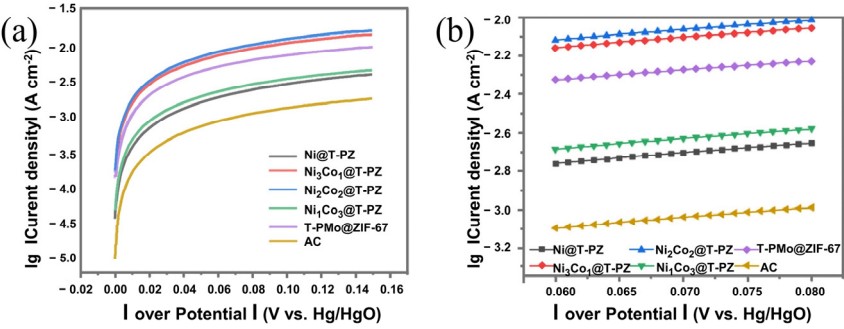

**Figure 3.** (**a**) Tafel curves of the $Ni_xCo_y$@T-PZ anodes; (**b**) Tafel curve fitting results for different $Ni_xCo_y$@T-PZ anodes (10 mL 3 M of KOH and 1 M of glucose solution).

2.1.4. Power Density (PD) and Polarization Curves (PC) of Different Anodes in Fuel Cells

In order to evaluate the performance of $Ni_2Co_2$@T-PZ in real alkaline glucose fuel cells, a whole-cell investigation was conducted through using carbon fabric as the air cathode, and catalyst-modified active carbon as the anode. Figure 4a shows the power density curves of the fuel cell with the different catalyst-modified anodes. The power densities of the fuel cell had a similar change trend, rising at first and then declining. The maximum power densities of the fuel cell with different anodes increased in the order of AC (9.060 W m$^{-2}$) < Ni@T-PZ (19.567 W m$^{-2}$) < $Ni_1Co_3$@T-PZ (21.515 W m$^{-2}$) < T-PMo@ZIF-67 (27.916 W m$^{-2}$) < $Ni_3Co_1$@T-PZ (31.774 W m$^{-2}$) < $Ni_2Co_2$@T-PZ (34.065 W m$^{-2}$). The power density of the fuel cell with $Ni_2Co_2$@T-PZ as the anode catalyst was always higher than that of the others with the maximum power density of 34.065 W m$^{-2}$, which was 1.22 times of that with T-PMo@ZIF-67 anode, 1.74 times of that with Ni@T-PZ anode, and 3.76 times of that with AC anode. Compared to other glucose fuel cells, our work has advantages in terms of performance and cost (Table 4).

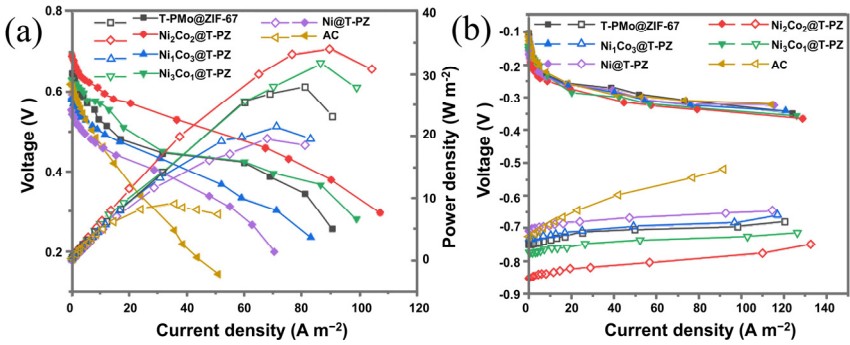

**Figure 4.** (**a**) Polarization and power density curves of the fuel cell equipped with different $Ni_xCo_y$@T-PZ doped anodes (The blank sign represents the power density curves of alkaline glucose fuel cells constructed with air cathodes and different catalytic anodes, and the filled sign represents the current density and open circuit voltage of the alkaline glucose fuel cells.); (**b**) Individual cathode and anode polarization curves (10 mL 3 M of KOH and 1 M of glucose solution).

**Table 4.** Performance of glucose fuel cells with different electrodes and catalysts.

| Anode | Cathode | Catalyst | Power Density | Reference |
|---|---|---|---|---|
| Ni | Al | Pt | 2 μW cm$^{-2}$; | [36] |
| Pt | Pt | Pt | 43 μW cm$^{-2}$; | [37] |
| PtPd/graphene | N-doped graphene oxide nanoribbons | Pt | 24.9 μW cm$^{-2}$; | [38] |
| Nanoporous gold | Pt/C | Pt/Bi | 8 mW cm$^{-2}$; | [39] |
| Activated carbon/Ni-foam | Activated carbon/Ni-foam | $Ni_2Co_2$@T-PMo@ZIF-67 | 3.4 mW cm$^{-2}$; | This work |

Figure 4b shows the polarization curves of the different electrodes. Because all the systems use the same air cathode, the changes of total voltage and power density should be attributed to different anodes. Among the six anodes, the potential of the AC anode decreased the fastest. The potentials of Ni@T-PZ and $Ni_1Co_3$@T-PZ anodes had a mild decrease compared to that of the AC anode. $Ni_2Co_2$@T-PZ anode had the most robust performance. This result is consistent with the LSV, EIS, and Tafel measurements.

### 2.2. Characterization of T-PMo@ZIF-67

### 2.2.1. Scanning Electron Microscope Analysis

SEM characterization of the catalyst was used to observe the microstructure of $Ni_2Co_2$@T-PZ (Figure 5a). The diameter of $Ni_2Co_2$@T-PZ particle was about 200 nm. The element mapping for $Ni_2Co_2$@T-PZ showed that C, Co, Ni, Mo, O, and P were uniformly distributed throughout the structure, which proved the successful doping of $Ni^{2+}$ in T-PMo@ZIF-67, and the distribution was uniform with no obvious agglomeration (Figure 5b). The energy dispersion spectra (EDS) of $Ni_2Co_2$@T-PZ catalyst indicated the molar ratio of Co, Mo, and Ni was 3.02:1:3.26 (Figure S2).

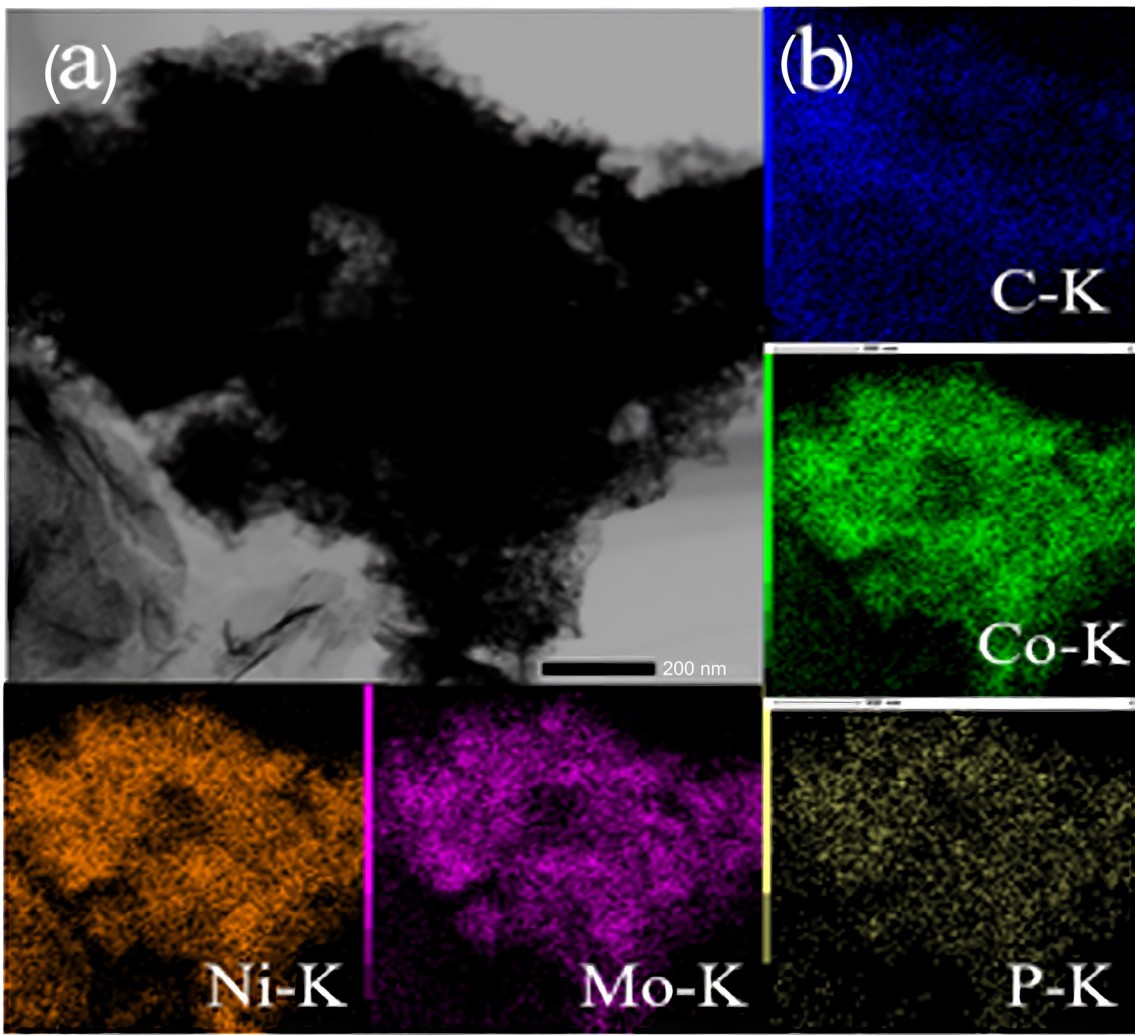

**Figure 5.** (**a**) SEM image of $Ni_2Co_2$@T-PZ. (**b**) Element diagram of $Ni_2Co_2$@T-PZ.

### 2.2.2. X-ray Diffraction Analysis

The XRD spectra of T-PMo@ZIF-67 and $Ni_2Co_2$@T-PZ (Figure 6) were compared, and the result showed that the peak size and peak area of the latter was remarkably increased, indicating a significant decrease in grain size. The presence of peaks at 36.6°, 43.3°, 45.2°,

and 50.6° represents the presence of CoNiO$_4$ and MoNiO$_4$, which indicates that Ni$^{2+}$ reacted with Co$^{2+}$ and Mo$^{2+}$ during the fabrication process. EDS and XRD results imply that Co$_3$O$_4$, CoMoO$_4$, CoNiO$_4$, and MoNiO$_4$ coexist in Ni$_2$Co$_2$@T-PZ.

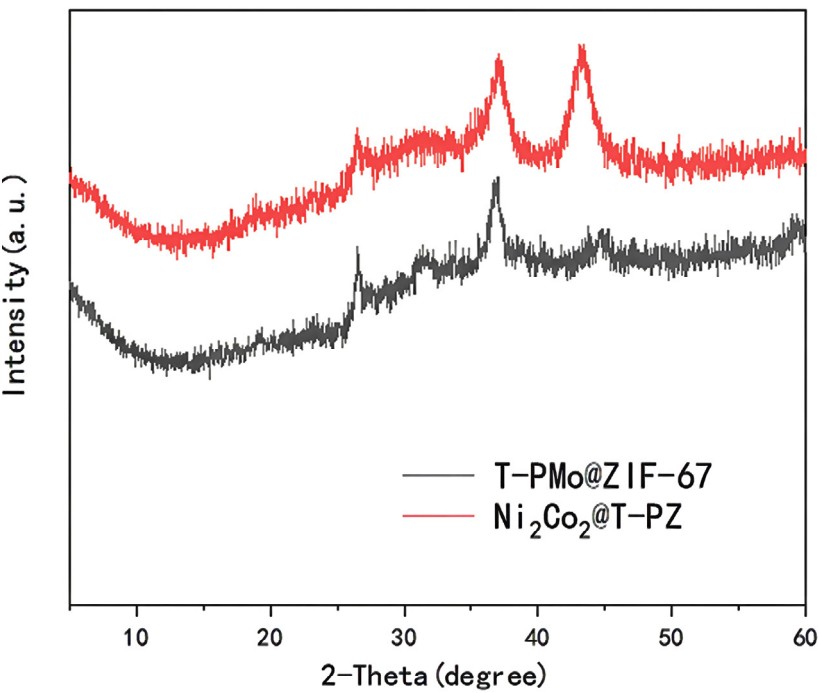

**Figure 6.** XRD pattern of T-PMo@ZIF-67 and Ni$_2$Co$_2$@ZIF-67.

### 2.2.3. X-ray Photoelectron Spectroscopy Analysis

The characterization of XPS is helpful when analyzing the elemental composition and valence distribution of Ni$_2$Co$_2$@T-PZ, so as to better understand the underlying catalytic mechanism. Figure 7a shows the XPS spectra of each element of Ni$_2$Co$_2$@T-PZ. The peaks centered at 855.10 eV, 796.06 eV, 530.14 eV, 284.95 eV, and 232.49 eV were Ni2p, Co2p, O1s, C1s, and Mo3d. The peaks centered at 284.77 eV, 286.28 eV, and 288.59 eV shown in Figure 7b represent C=C/C-C, C=C/C-C, and C=O for C1s signals. In Figure 7c, it can be seen there were two splitting slits at 235.34 eV and 232.22 eV in the XPS pattern of Mo3d, and the column width was 3.12 eV, which can be ascribed to Mo$^{6+}$. The spectrum of Co2p was mainly derived from the splitting of two orbitals, which were spin orbital double peaks with a column width of about 15 eV (Figure 7d). The spin orbital double peaks at 780 eV and 795 eV represented the Co2p$_{3/2}$ and Co2p$_{1/2}$ levels of the Co ion, respectively. The satellite peaks could be used to identify the oxidation state of Co ions. The energy gap between satellite peaks and the main peaks was 6 eV and 8 eV respectively, indicating that Co$^{2+}$ and Co$^{3+}$ co-exist in the sample. Figure 7e shows the XPS spectrum of Ni2p. The peaks at the binding energy of 853.97 eV and 860.98 eV was the characteristic peak of Ni$^{3+}$, and the double peak at the binding energy of 855.87 eV and 863.14 eV was the characteristic peak of Ni$^{2+}$. This result indicates that Ni$^{3+}$ and Ni$^{2+}$ coexist in the sample.

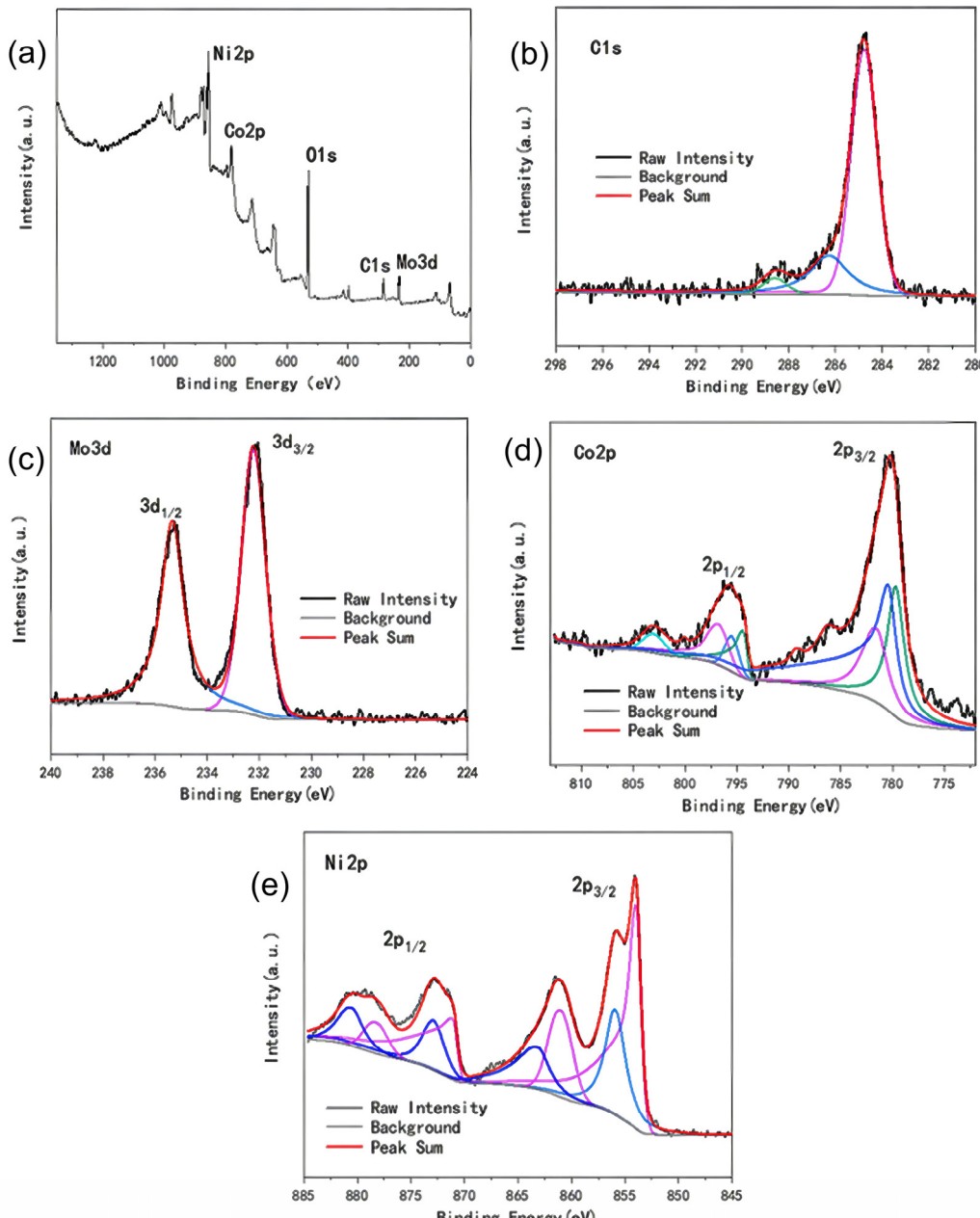

**Figure 7.** (**a**) Full spectrum of XPS for Ni2Co2@ZIF-67; (**b**) C1s spectra; (**c**) Mo3d spectra; (**d**) Co2p spectra; (**e**) Ni2p spectra.

### 2.3. Catalytic Mechanism

SEM, XPS, XRD, and electrochemical characterization results can be used to infer the catalytic mechanism of glucose oxidation by Ni$_2$Co$_2$@T-PZ in alkaline environment. The catalytic process can involve the following steps [40–45]:

$$24\,Co^{2+} - 24\,e^- \rightarrow 24\,Co^{3+} \tag{1}$$

$$C_6H_{12}O_6 + 36\,OH^- + 24\,Co^{3+} \rightarrow 6CO_3{}^{2-} + 24\,Co^{2+} + 24\,H_2O \tag{2}$$

$$24\,Ni^{2+} - 24\,e^- \rightarrow 24\,Ni^{3+} \tag{3}$$

$$C_6H_{12}O_6 + 36\,OH^- + 24\,Ni^{3+} \rightarrow 6CO_3{}^{2-} + 24\,Ni^{2+} + 24\,H_2O \tag{4}$$

$$6\,Mo^{6+} + 12\,e^- \rightarrow 6\,Mo^{4+} \tag{5}$$

$$6\ Mo^{4+} + 12\ e^{-} \rightarrow 6\ Mo^{2+} \tag{6}$$

$$C_6H_{12}O_6 + 36\ OH^{-} + 6\ Mo^{6+} \rightarrow 6CO_3^{2-} + 6\ Mo^{2+} + 24\ H_2O \tag{7}$$

$$Mo^{6+} + 2\ Co^{2+} \rightarrow Mo^{4+} + 2\ Co^{3+} \tag{8}$$

$$Mo^{6+} + 2\ Ni^{2+} \rightarrow Mo^{4+} + 2\ Ni^{3+} \tag{9}$$

$Co^{3+}$ and $Ni^{3+}$ generated from $Co^{2+}$ and $Ni^{2+}$ have a high oxidative state, which can accelerate the oxidation process of glucose under alkaline environment. $Mo^{6+}$ has great tendency to acquire electrons from neighbor molecules. Firstly, it can react with $Co^{2+}$ and $Ni^{2+}$ to produce $Co^{3+}$ and $Ni^{3+}$. In addition, it can directly capture electrons from glucose and be reduced to $Mo^{2+}$. Figure 8 depicts the possible reaction processes that $Ni_2Co_2@T$-PZ involved in the oxidation of glucose. There may exist a synergistic effect between $Co^{2+}$, $Ni^{2+}$, and $Mo^{6+}$: with the good oxidizing property of $Mo^{6+}$, $Co^{3+}$, and $Ni^{3+}$ being easily obtained, and $Co^{3+}$ and $Ni^{3+}$ being re-reduced to $Co^{2+}$ and $Ni^{2+}$ by oxidizing glucose. Meanwhile, there is a synergistic effect between reactions (1) and (3), $Ni^{3+}$ has the ability to capture electrons from glucose, and $Co^{2+}$ can play a facilitating role. The resulting $Mo^{2+}$ is oxidized to $Mo^{6+}$ via the cathodic transfer of electrons, resulting in catalyst recycling [46,47]. In addition, there is a good synergistic effect between POM and ZIF-67, so $Ni_2Co_2@T$-PZ catalyst has a high specific surface area and abundant active sites, and also has excellent diffusion efficiency. The $Ni_2Co_2@T$-PZ catalyst showed excellent catalytic performance for glucose oxidation [48,49].

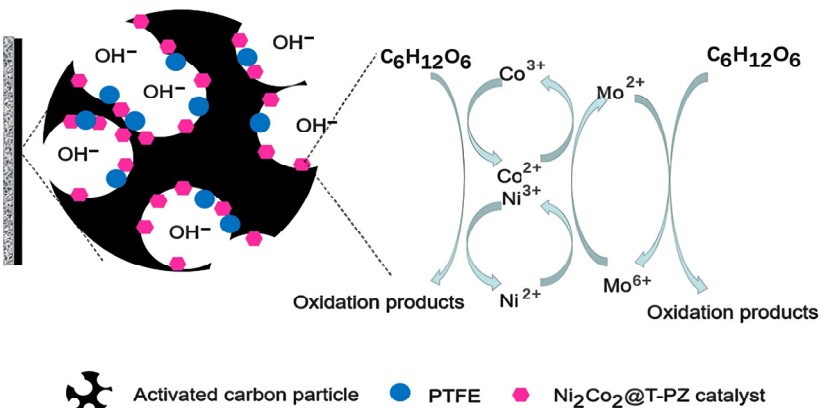

**Figure 8.** Mechanism diagram of $Ni_2Co_2@T$-PZ catalyzed glucose oxidation.

## 3. Materials and Methods

### 3.1. Materials

Nickel foam was obtained from Yilongsheng Energy Technology Co., Ltd. (Suzhou, China). Activated Carbon (AC) powder (YEC-8A) was purchased from Yihuan Carbon Co., Ltd. (Fuzhou, China). A 60 wt% PTFE solution was purchased from Hesen Inc. (Shanghai, China). Phosphomolybdic acid hydrate (PMo12) and 2-methylimidazole were purchased from Kemiou (Tianjin, China). Glucose, KOH, and all the other chemicals were analytically pure. Deionized water (DI) was used as the solvent for all the solutions.

### 3.2. Characterization

Electrochemical characterization was carried out on a CHI-660E electrochemical workstation (Chenhua Instrument Co., Ltd., Shanghai, China). The microstructure of the samples was analyzed on a SEM (S4800, Hitachi High-tech Corporation, Tokyo, Japan) and a TEM (Tecnai G2 F20, IKA, Königswinter, Germany). X-ray diffraction (XRD) patterns were measured by a D/MX-IIIA X-ray diffractometer (Rigaku, Japan). XPS analysis was carried out on an Escalab 250Xi X-ray photoelectron spectrometer (Thermo Fisher Scientific, Waltham, MA, USA).

### 3.3. Preparation of Electrodes

#### 3.3.1. Preparation of XPMo@ZIF-67 and Ni$_x$Co$_y$@PZ

An amount of 2.5 mM cobalt nitrate, together with 2.5 mM different transitional metal salts (copper nitrate, ferric nitrate, nickel nitrate or zinc nitrate), was added into 25 mL of methanol. After fully mixing, 10 mL of 30 mM phosphomolybdate aqueous solution was added into the bottle to obtain solution A (denoted as A1, A2, A3, and A4, respectively). Solution B was prepared by adding 20 mM 2-methylimidazole into 25 mL methanol. Solutions C was obtained by quickly mixing solution A with solution B (denoted as C1, C2, C3, and C4, respectively). Then, the mixture was heated in a hydrothermal reactor (10 h, 120 °C). Afterwards, purple precipitates were obtained. The obtained precipitates were filtered, and then were washed with ethyl alcohol. The precipitates were dried in a 60 °C oven overnight to obtain CuPMo@ZIF-67, FePMo@ZIF-67, NiPMo@ZIF-67, and ZnPMo@ZIF-67, respectively.

To prepare Ni$_x$Co$_y$@PZ, the Ni$^{2+}$/Co$^{2+}$ ratio was modified during the preparation of NiPMo@ZIF-67. The amounts of 0.6 mM nickel nitrate and 1.9 mM cobalt nitrate, 1.25 mM nickel nitrate and 1.25 mM cobalt nitrate, 1.9 mM nickel nitrate and 0.6 mM cobalt nitrate, and 2.5 mM nickel nitrate were dissolved in 25 mL methanol solution, respectively to obtain Ni$_1$Co$_3$@PZ, Ni$_2$Co$_2$@PZ, Ni$_3$Co$_1$@PZ, and Ni@PZ, respectively.

#### 3.3.2. Preparation of T-XPMo@ZIF-67 and Ni$_x$Co$_y$@T-PZ

XPMo@ZIF-67 was placed into the Muffle furnace and slowly heated to 500 °C for 2 h. After it cooled down to room temperature, the synthesized products were washed with ethyl alcohol, and the precipitates were dried in a 60 °C oven to obtain T-NiPMo@ZIF-67, T-CuPMo@ZIF-67, T-FePMo@ZIF-67, T-ZnPMo@ZIF-67, respectively. The whole procedure of the preparation of T-NiPMo@ZIF-67, as well as the schematic diagram, were depicted in Figure 9a,b, respectively.

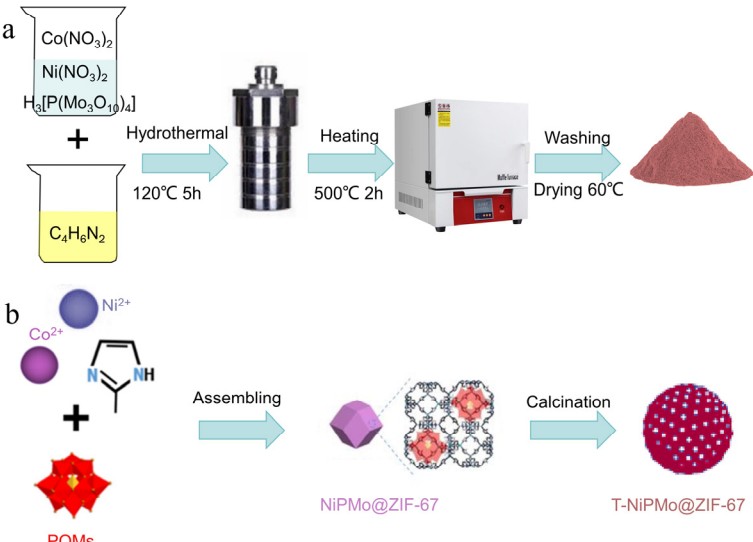

**Figure 9.** (**a**) The flow chart of the preparation of T-NiPMo@ZIF-67; (**b**) The schematic diagram of the preparation of T-NiPMo@ZIF-67.

Ni$_x$Co$_y$@T-PZ was prepared using the same procedure from Ni$_x$Co$_y$@PZ, and was denoted as Ni$_1$Co$_3$@T-PZ, Ni$_2$Co$_2$@T-PZ, Ni$_3$Co$_1$@T-PZ, and Ni@T-PZ, respectively.

#### 3.3.3. Preparation of T-XPMo@ZIF-67 and Ni$_x$Co$_y$@PZ Modified Anode

The T-XPMo@ZIF-67 and Ni$_x$Co$_y$@PZ modified anodes were prepared by rolling method as previously described [23]. The detail procedure is as follows:

(1) An amount of 0.5 g of activated carbon was mixed with T-XPMo@ZIF-67/$Ni_xCo_y$@PZ in a ratio of 1:0.01~1:0.05 by using anhydrous ethanol as solvent. The obtained mixture was placed in an ultrasonic cleaner with ultrasonic shaking and constant stirring for 40 min;

(2) An amount of 0.25 g PTFE emulsion (60 wt%) was slowly added, and stirred continuously for 40 min;

(3) The mixture was placed in a thermostatic water bath at 70 °C, and stirred continuously;

(4) The catalyst modified anode with a thickness of 2~4 mm was obtained by rolling the mixture onto a Ni foam disc (diameter: 36 mm).

The blank control anode was prepared by using the same procedure with 0.5 g activated carbon powder.

### 3.4. Experimental Methods

The fuel cell was mainly made of polymethyl methylbenzoate (PMMA), with carbon cloth and nickel foam seals at both ends as the cathode and anode. An amount of 10 mL of 1 M glucose in 3 M KOH solution was used as substrate. The water used in the experiment was deionized water. A three-electrode system was employed in the electrochemical experiment. The reference electrode was saturated HgO electrode, the counter electrode was carbon cloth, and the working electrode was catalyst modified activated carbon/nickel foam. For the characterization of whole-cell performance (polarization curves and power density curves of the fuel cell), the fuel cell was assembled with a $Ni_xCo_y$@T-PZ doped anode and an active carbon air-cathode [44,45]. When placing the air cathode, we made sure that the catalytic layer of the air cathode faced the inner cavity of the fuel cell. The catalyst modified anode and the air cathode were pressed tightly with the rubber gasket to make them fully in contact with the nickel wire. After assembling the fuel cell, we checked whether there was water leakage and air tightness. Then, the glucose and KOH were injected into the inner cavity of the fuel cell from the inlet port on the side of the fuel cell, and the nitrogen was quickly vented for 10 min, so that the residual oxygen in the fuel cell chamber was discharged. The outlet was water-sealed during the exhaust process, and continuous small bubbles could be observed. After passing nitrogen, we sealed the air inlet and outlet of the fuel cell to prevent oxygen in the air from entering the fuel cell directly through these two ports. The temperature of the test environment was 23 °C ± 2 °C. The change of fuel cell voltage was measured in real time, and various electrochemical parameters were measured when the open circuit voltage was stable.

### 4. Conclusions

In this paper, a series of efficient catalysts for glucose oxidation were prepared by doping transitional metals into nanohybrids of PMo and ZIF-67. The doping of $Ni^{2+}$ could significantly improve the electrochemical performance of T-PMo@ZIF-67. Among them, $Ni_2Co_2$@T-PZ showed the best performance. The maximum power density of the glucose cell doped with $Ni_2Co_2$@T-PZ catalyst reached 34.065 W $m^{-2}$, which was much higher than that of conventional cobalt-molybdenum-nickel oxides. The characterization results by SEM, XPS, and XRD indicate that $Ni_2Co_2$@T-PZ is a nanohybrid, being composed of $Co_3O_4$, $CoMoO_4$, $CoNiO_4$, and $MoNiO_4$. $Ni_2Co_2$@T-PZ maintains the hollow structure of ZIF-67, and there is a synergistic effect between $Co^{2+}$, $Ni^{2+}$, and $Mo^{6+}$ for glucose oxidation. This work has considerable application prospects in the field of electrochemistry.

**Supplementary Materials:** The following supporting information can be downloaded at: https://www.mdpi.com/article/10.3390/catal14010019/s1. Figure S1: Equivalent circuit diagram; Figure S2: EDS image of $Ni_2Co_2$@T-PZ.

**Author Contributions:** Writing—original draft preparation, S.J. and N.K.; methodology, Y.Z. and Y.L.; data curation, M.L. and B.M.; conceptualization, N.K., X.Z. and X.L.; writing—review and editing, P.Z. and X.L. All authors have read and agreed to the published version of the manuscript.

**Funding:** This work was partially financially supported by the National Natural Science Foundation of China (No. 42377380) and the National Key R&D Program of China (Grant No. 2019YFC1407800).

**Data Availability Statement:** The data presented in this study are available on request from the corresponding author.

**Conflicts of Interest:** Authors have no conflict of interest to declare.

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
