# Peer review of "Boosted Catalytic Performance of Ni2Co2@T-PMo@ZIF-67 for Glucose Oxidation in a Direct-Glucose Fuel Cell"

_catalysts, doi:10.3390/catal14010019_

Round 1

Reviewer 1 Report

Comments and Suggestions for Authors

In the present manuscript titled: Boosted Catalytic Performance of Ni2Co2@T-PMo@ZIF-67 for Glucose Oxidation in a Direct-Glucose Fuel Cell (by the authors: Jiao, S.; Kang, N.; Liu, M.; Zhang, Y.; Li, Y.; Maryam, B.; Zhang, X.; Zhang, P.), catalysts for glucose oxidation were prepared and characterized with the aid of physical and electrochemical techniques. Therefore Ni2+ was doped into a PMo@ZIF-67 in order to create NixCoy@T-PMo@ZIF-67 catalyst. Several other metal ions such as Fe3+, Cu2+, or Zn2+ were also doped into a PMo@ZIF-67 and further tested in glucose oxidation in an alkaline solution. The results are interesting, but an additional explanation of the experimental findings is necessary.

Some considerations and questions are listed below:

o   In the Abstract: molecular organic frameworks (MOFs). Molecular or metal?

o   From Fig. 1 and the results of linear sweep voltammetry the doping of Ni2+ in the catalyst increased the oxidation current of the glucose. What is the appropriate amount of Ni2+? Highlight the role of nickel.

o   Synergism between Co2+ and Ni2+ was achieved according to Fig. 1b, but it is not clear what is the optimal Co2+/Ni2+ ratio as a molar ratio. What happens when a catalyst consists of more Co2+ or more Ni2+?

o   In Figure caption add the glucose concentration and the solution.

o   There is a lack of explanation for the results in Fig. 3a.

o   What is the role of Fe3+, Cu2+ or Zn2+ in the catalyst? Some explanation is necessary.

o   In Fig. 5 add the CVs of NixCox@T-PZ catalyst in the absence and in the presence of glucose in solution.

o   What is the facilitating role of Co2+? Some explanation supported by literature is necessary.

o   Compare the performance parameters for glucose oxidation on various electrodes with literature findings.

o   In the experimental part check the unit of scan rate.

Author Response

Reviewer #1:

In the present manuscript titled: Boosted Catalytic Performance of Ni2Co2@T-PMo@ZIF-67 for Glucose Oxidation in a Direct-Glucose Fuel Cell (by the authors: Jiao, S.; Kang, N.; Liu, M.; Zhang, Y.; Li, Y.; Maryam, B.; Zhang, X.; Zhang, P.), catalysts for glucose oxidation were prepared and characterized with the aid of physical and electrochemical techniques. Therefore Ni2+ was doped into a PMo@ZIF-67 in order to create NixCoy@T-PMo@ZIF-67 catalyst. Several other metal ions such as Fe3+, Cu2+, or Zn2+ were also doped into a PMo@ZIF-67 and further tested in glucose oxidation in an alkaline solution. The results are interesting, but an additional explanation of the experimental findings is necessary.

Some considerations and questions are listed below:

Comment 1. In the Abstract: molecular organic frameworks (MOFs). Molecular or metal?

Response:

Thanks for your comment. It should be metal-organic frameworks (MOFs).  We have corrected it in the revised manuscript. 

Comment 2. From Fig. 1 and the results of linear sweep voltammetry the doping of Ni2+ in the catalyst increased the oxidation current of the glucose. What is the appropriate amount of Ni2+? Highlight the role of nickel.

Response:

Thanks for your suggestion. Fig.1a shows the doping of Ni2+ in the catalyst significantly increased the oxidation current of the glucose. The appropriate amount of Ni2+ was indicated in Fig. 1b. The appropriate amount of Ni2+ was 1.25 mM Ni2+, with which the catalyst showed the best performance. In the revised manuscript, we make it clear and highlight the role of nickel.

Comment 3. Synergism between Co2+ and Ni2+ was achieved according to Fig. 1b, but it is not clear what is the optimal Co2+/Ni2+ ratio as a molar ratio. What happens when a catalyst consists of more Co2+ or more Ni2+?

Response:

Thank you for your suggestion. In the revised manuscript, we make it clear that the optimal Co2+/Ni2+ ratio as a molar ratio was 1:1, and when the catalyst is composed of more Co2+ or more Ni2+, the catalytic performance of the catalyst will deteriorate.

“Figure 1b shows the LSV measurements of the catalysts with different Co2+/Ni2+ ratios. It can be seen that the four modified anode curves are above the blank anode curves, and the maximum current density relationship is AC < Ni@T-PZ < Ni1Co3@T-PZ < Ni3Co1@T-PZ < Ni2Co2@T-PZ. This result further confirms that doping with appropriate amount of Ni2+ (1.25 mM) can promote the oxidation of glucose and the optimal Co2+/Ni2+ ratio as a molar ratio was 1:1. Co2+ and Ni2+ may have a synergistic effect on the catalytic oxidation capacity of glucose. Therefore, the LSV curves of Ni3Co1@T-PZ and Ni2Co2@T-PZ are higher than that of Ni@T-PZ, which Ni2+ was added alone. However, when the concentration of either metal ion is higher, the catalytic performance of the catalyst will deteriorate.”

Comment 4. In Figure caption add the glucose concentration and the solution.

Response:

Thank you for your suggestions. The glucose concentration and solution are now added in the Figure caption.

Comment 5. There is a lack of explanation for the results in Fig. 3a.

Response:

Thanks for your suggestion. Fig. 3b are obtained from Fig. 3a. In the revised paper, we explain it clearly.

 “The Tafel curve is a section of the strong polarization region of the general polarization curve. For the simpler electron transfer process, the Tafel curve can be used for analysis, and the exchange current density can be obtained by using the linear part of the Tafel curve to intersect the axis through the extension line, so as to calculate the number of electrons transferred in the electrochemical process. Figure 3a shows the Tafel curves of different catalyst modified anodes, and Figure 3b shows the linear parts of the Tafel curves and the fitting curves for different NixCoy@T-PZ anodes.” 

Comment 6. What is the role of Fe3+, Cu2+ or Zn2+ in the catalyst? Some explanation is necessary.

Response:

Thanks for your advice. We have explained this in the paper.

“The oxidation current increases in the order of T-ZnPMo@ZIF-67 < AC < T-FePMo@ZIF-67 < T-CuPMo@ZIF-67 < T-PMo@ZIF-67 < T-NiPMo@ZIF-67. This indicates that Zn2+ remarkably impair the catalytic activity of T-PMo@ZIF-67. It can be inferred that Zn2+ has a lower catalytic activity towards glucose oxidation. In addition, it will compete with Co2+ for bridging linker (2-methylimidazole) to form ZIF-8. Although Cu2+ and Fe2+ can promote the glucose oxidation reaction, the catalytic performances of T-CuPMo@ZIF-67 and T-FePMo@ZIF-67 were not as high as that of T-PMo@ZIF-67. The doping of Ni2+ in the catalyst significantly increased the oxidation current of the glucose”. 

Comment 7. In Fig. 5 add the CVs of NixCox@T-PZ catalyst in the absence and in the presence of glucose in solution.

Response:

Thanks for your suggestion. As reviewer 2 suggested we have removed this figure.

Comment 8. What is the facilitating role of Co2+? Some explanation supported by literature is necessary.

Response:

Thanks for your advice. We have explained the facilitating role of Co2+ and provided literature support.

“Co3+ and Ni3+ generated from Co2+ and Ni2+ have a high oxidative state, which can accelerate the oxidation process of glucose under alkaline environment. Mo6+ has great tendency to acquire electrons from neighbor molecules. Firstly, it can react with Co2+ and Ni2+ to produce Co3+ and Ni3+. In addition, it can directly capture electrons from glucose and be reduced to Mo2+. Figure 9 depicts the possible reaction processes that Ni2Co2@T-PZ involved in the oxidation of glucose. There may exist a synergistic effect between Co2+, Ni2+ and Mo6+: with the good oxidizing property of Mo6+, Co3+ and Ni3+ can be easily obtained, and Co3+ and Ni3+ can be re-reduced to Co2+ and Ni2+ by oxidizing glucose. Meanwhile, there is a synergistic effect between reactions (1) and (3), Ni3+ has the ability to capture electrons from glucose, and Co2+ can play a facilitating role. The generated Mo2+ is oxidized to Mo6+ by transferring electrons through the cathode, and the electrons are transferred from the cathode to the external circuit, thus realizing the recycling of the catalyst, ensuring the stability of the battery and improving the performance of the battery [46,47]”. 

Comment 9. Compare the performance parameters for glucose oxidation on various electrodes with literature findings.

Response:

Thanks for your suggestion, we have revised the paper.

Table 4. Performance of glucose fuel cells with different electrodes and catalysts. 

Anode

Cathode

Catalyst

Power density

Reference

Ni

Al

Pt

2 μW cm⁻²

[36]

Pt

Pt

Pt

43 μW cm⁻²

[37]

PtPd/graphene

N-doped graphene oxide nanoribbons

Pt

24.9 µW cm⁻²

[38]

Nanoporous gold

Pt/C

Pt/Bi

8 mW cm⁻²

[39]

Activated carbon/Ni-foam

Activated carbon/Ni-foam

Ni2Co2@T-PMo@ZIF-67

3.4 mW cm⁻²

This work

Comment 10. In the experimental part check the unit of scan rate.

Response:

Thanks for your suggestion, we have revised the paper.

Change 0.1 mV to 0.1 mV s-1;

Change 1 mV to 1 mV s-1.

Reviewer 2 Report

Comments and Suggestions for Authors

The manuscript needs substantial improvement in order to be cosidered for publication

1. Please include some more sentences on the glucose fuel cell and its use.

2. Please explain the E_initial values (open circuit E_oc) in Fig. 1

3. The shape of the LSV diagrams is somehow strange. EIS of the cell?

4. We cannot observe a straight line in Fig. 2, menaing no diffusion part is observed. Further you give the used circuit for fitting the impedance results as R(RC/RC). In this no Warburg impedance is included, neither Rt and Rd.

How did you obtain these values you present in Table 1

5. The perfect linearity shown in Fig. 3b is not obvious from Fig. 3a. Please explain.

6. Please give units for Tafel and I in Table 2 

7. Please explain the proposed mechanism in p.9. Is it party chemical or electrochemical redox of the involved cations?

8. You mentioned than this CAN be the mechanism. How sure are you the it is true in your case ? Do the references 33, 34 and 35 truly support the proposed mechanism??

9. Please explain the UI-Curves in Fig. 4. Why do you believe that the max. power should be  calculated at the highest current although the corresponding voltage is in the diffusion limited part of the curve?

in Fig. 5 I do not observe any peaks for a real redox system but rather capacitive behavior of the system. Please explain or reconsider your results and description.

Comments on the Quality of English Language

Only minor English polishing should be done

Author Response

Reviewer #2:

The manuscript needs substantial improvement in order to be considered for publication.

Comment 1. Please include some more sentences on the glucose fuel cell and its use.

Response:

Thanks for your suggestion, we have revised the paper.

In Section 1, the following sentences are added:

“Glucose is the most abundant monosaccharide in nature, which is cheap and easy to obtain, non-volatile, safe and environmentally friendly, green and non-toxic [22]. glucose fuel cells can be divided into two categories according to the operating environment: External fuel cells and implantable fuel cells. In the case of external operation, glucose fuel cells can directly use agricultural waste as an energy source for power generation, which has the advantages of simplicity, safety, environmental protection and wide fuel sources. Implantable fuel cells use glucose in body fluids such as interstitial fluid, tears, blood, cerebrospinal fluid to power the corresponding implanted devices, and have broad application prospects in implantable medical devices such as cardiac pacemakers and drug delivery pumps [23, 24].” 

Comment 2. Please explain the E_initial values (open circuit E_oc) in Fig. 1.

Response:

Thanks for your advice. The configuration is half cell, therefore the initial points are called open potentials. We have explained the open potentials in Fig. 1 in the revised manuscript.

The oxidation current increases in the order of T-ZnPMo@ZIF-67 < AC < T-FePMo@ZIF-67 < T-CuPMo@ZIF-67 < T-PMo@ZIF-67 < T-NiPMo@ZIF-67. The open potentials of the different anodes follow the same trend. T-NiPMo@ZIF-67 has the lowest open potential (-0.801 V vs Hg/HgO), and T-ZnPMo@ZIF-67 had the highest open potential (0.637 V vs Hg/HgO).

Comment 3. The shape of the LSV diagrams is somehow strange. EIS of the cell?

Response:

Thanks for your suggestion, we have carefully checked the LSV diagram and EIS diagram.

In some papers, the LSV diagram takes the current density as the horizontal coordinate and the voltage as the vertical coordinate. In our paper, potential is taken as the horizontal coordinate and current density as the vertical coordinate.

Comment 4. We cannot observe a straight line in Fig. 2, menaing no diffusion part is observed.  Further you give the used circuit for fitting the impedance results as R(RC/RC).  In this no Warburg impedance is included, neither Rt and Rd.

How did you obtain these values you present in Table 1

Response:

Thanks for your suggestion. The R(RC)(RC) circuit contains three Resistances, and we refer them as Rs, Rct, and Rd, respectively. Rt = Rs + Rct + Rd. We have supplemented the equivalent circuit diagram in the supporting material. The method of calculating the values in Table 1 is also given.

Figure S1. Equivalent circuit diagram.

The results were fitted according to ZSimpWin. The equivalent circuit was R(CR)(CR) as shown in Figure S1. Ohmic resistance (Rs), layer capacitance (Cdl), charge transfer resistance (Rct), void adsorption capacitance (Cad) and diffusion resistance (Rd) can be calculated by fitting, and the total resistance of the battery can be calculated. In addition, the total ohmic resistance (Rt) in the fuel cell anode system represents the sum of the resistance values of Rs, Rct, and Rd.

Comment 5. The perfect linearity shown in Fig. 3b is not obvious from Fig. 3a.  Please explain.

Response:

Thanks for your suggestion, we have carefully examined Figures 3a and 3b. In terms of vision, the two figures do not match well because the horizontal coordinate data range of the two figures is different. The horizontal coordinate data range of Figure 3a is from -0.02 to 0.16, and the horizontal coordinate data range of Figure 3b is from 0.0575 to 0.0.0825.

Comment 6. Please give units for Tafel and I in Table 2. 

Response:

Thanks for your suggestions, which helps improve the level of our paper. We have revised the paper and given the units of Tafel and I0.

Table 3. Tafel curve fitting results. 

Fitting Equation

R2

Tafel

 (mV dec-1)

10−4I0

(A cm-2)

AC

y=5.37143 × -3.41673

0.9983

186.1702

3.831

Ni@T-PZ

y=5.41948 × -3.08181

0.99818

184.5195

8.283

Ni3Co1@T-PZ

y=5.05714 × -2.45716

0.99876

197.7402

34.901

Ni2Co2@T-PZ

y=5.14286 × -2.42154

0.99758

194.4443

37.884

Ni1Co3@T-PZ

y=5.21818 × -2.99496

0.99807

191.6377

10.117

T-PMo@ZIF-67

y=4.82208 × -2.61037

0.99766

207.3794

24.526

Comment 7. Please explain the proposed mechanism in p.9.  Is it party chemical or electrochemical redox of the involved cations?

Response:

Thanks for your suggestion, we have revised the paper. Mechanisms include REDOX of related cations and synergistic action of POM and ZIF-67.

Co3+ and Ni3+ generated from Co2+ and Ni2+ have a high oxidative state, which can accelerate the oxidation process of glucose under alkaline environment. Mo6+ has great tendency to acquire electrons from neighbor molecules. Firstly, it can react with Co2+ and Ni2+ to produce Co3+ and Ni3+. In addition, it can directly capture electrons from glucose and be reduced to Mo2+. Figure 9 depicts the possible reaction processes that Ni2Co2@T-PZ involved in the oxidation of glucose. There may exist a synergistic effect between Co2+, Ni2+ and Mo6+: with the good oxidizing property of Mo6+, Co3+ and Ni3+ can be easily obtained, and Co3+ and Ni3+ can be re-reduced to Co2+ and Ni2+ by oxidizing glucose. Meanwhile, there is a synergistic effect between reactions (1) and (3), Ni3+ has the ability to capture electrons from glucose, and Co2+ can play a facilitating role. The resulting Mo2+ is oxidized to Mo6+ by cathodic transfer of electrons, resulting in catalyst recycling [46],[47]. In addition, there is a good synergistic effect between POM and ZIF-67, so Ni2Co2@T-PZ catalyst has a high specific surface area and abundant surface active sites, and has excellent diffusion efficiency. The Ni2Co2@T-PZ catalyst showed excellent catalytic performance for glucose oxidation [48],[49].

Comment 8. You mentioned than this CAN be the mechanism.  How sure are you the it is true in your case?  Do the references 33, 34 and 35 truly support the proposed mechanism??

Response:

Thanks for your suggestions, we proposed this mechanism based on previous publications. Verification of this mechanism need more work to be done, which will be the focus of our next work.  In the revised the paper, we provided more references to support the proposed mechanism.

43.Yang, X.-L.; Ye, Y.-S.; Wang, Z.-M.; Zhang, Z.-H.; Zhao, Y.-L.; Yang, F.; Zhu, Z.-Y.; Wei, T.  POM-Based MOF-Derived Co3O4/CoMoO4Nanohybrids as Anodes for High-Performance Lithium-Ion Batteries, Acs Omega. 2020, 5, 26230-26236.

44.Gao, M.; Liu, X.; Irfan, M.; Shi, J.; Wang, X.; Zhang, P.  Nickle-cobalt composite catalyst-modified activated carbon anode for direct glucose alkaline fuel cell, Int J Hydrogen Energ. 2018, 43, 1805-1815.

45.Irfan, M.; Khan, I. U.; Wang, J.; Li, Y.; Liu, X.  3D porous nanostructured Ni3N-Co3N as a robust electrode material for glucose fuel cell, Rsc Adv. 2020, 10, 6444-6451.

Comment 9.  Please explain the UI-Curves in Fig. 4.  Why do you believe that the max. power should be calculated at the highest current although the corresponding voltage is in the diffusion limited part of the curve?

Response:

Thanks for your comment. As is shown in Fig.4, the peak power densities were not calculated at the highest current densities. We didn’t measure the maximum current, and we stopped the measurements when the calculated power density became decline. In the revised manuscript, we have provided the calculation method of power density in the supplementary material.

When the open circuit voltage was stable, the resistance box was connected to the resistor in parallel, and the resistance size of the resistance box was adjusted from 9000 Ω to 3 Ω. After the resistance was adjusted by the resistance box, the multimeter data was recorded after the voltage was stable. The power density of the battery and the polarization data of the anode and cathode were calculated according to the formulas (1) and (2), and then the picture was drawn with Origin.

I=U/RS

(1)

P=U×I

(2)

Where I is the current density (A m-2), U is the voltage (V), R is the resistance box resistance (Ω), S is the electrode area, and P is the power density (W m-2).

Comment 10. In Fig. 5 I do not observe any peaks for a real redox system but rather capacitive behavior of the system.  Please explain or reconsider your results and description.

Response:

Thanks for your suggestion, we took a closer look at Figure 5. We have revised the paper to remove Figure 5.

Round 2

Reviewer 1 Report

Comments and Suggestions for Authors

The authors responded to all the questions and the manuscript is now accepted.

Author Response

Thank you for your advices and support

Reviewer 2 Report

Comments and Suggestions for Authors

I still disagree with the interpretation of the EIS diagrams!

There is no Warburg impedance (Straight line with 45 ° at low frequences) detected!

You cannot name a resistance "Diffusion resistance" and just fit the data. Please read carefully original literature about Warburg impedance and correct the manuscript accordingly!

Author Response

Response to the reviewers' comments:

Reviewer #2:

Comment 1. We cannot observe a straight line in Fig. 2, menaing no diffusion part is observed.  Further you give the used circuit for fitting the impedance results as R(RC/RC). In this no Warburg impedance is included, neither Rt and Rd.

How did you obtain these values you present in Table 1

Response:

Thanks for your comment. We are sorry for our vague description on the diffusion resistance. The assumptions of the classical Warburg impedance are very strict, that is, semi-infinite, planar, purely concentration-gradient-driven, Fickian diffusion of neutral species in the dilute limit. In reality, we are, however, frequently encountered with ‘non-ideal’ diffusion processes, including but not limited to, diffusion in bounded or irregular space, diffusion of charged species which is coupled with migration and/or convection, and multi-scale and multi-phase diffusion in porous electrodes [1-3]. In consequence, it shall not be surprising that non-idealities of the diffusion impedance, including arced curves, are frequently observed in experiments. In this work, the low frequency region of the Nyquist plot is not a straight line but an arced curve. We found the EIS data can be well fitted by the circuit R(CR)(CR), better than R(CR)(C(RW))[4]. We have supplemented the labelled equivalent circuit diagram in the supporting material. The calculation methods of each value (Rs, Rd, and Rct) in Table 1 are also given. As to Rt, it is the sum of the resistance values of Rs, Rct, and Rd.

Figure S1. Equivalent circuit diagram.

The results were fitted according to ZSimpWin. The equivalent circuit was R(CR)(CR) as shown in Figure S1. Ohmic resistance (Rs), layer capacitance (Cdl), charge transfer resistance (Rct), void adsorption capacitance (Cad) and diffusion resistance (Rd) can be calculated by data fitting. In addition, the total ohmic resistance (Rt) in the fuel cell anode system represents the sum of the resistance values of Rs, Rct, and Rd.

[1] Jun Huang. Diffusion impedance of electroactive materials, electrolytic solutions and porous electrodes: Warburg impedance and beyond. Electrochimica Acta. 2018; 281:170-188

[2] Qiu X-Y, Zhuang Q-C, Zhang Q-Q, Cao R, Ying P-Z, Qiang Y-H, et al. Electrochemical and electronic properties of LiCoO2 cathode investigated by galvanostatic cycling and EIS. Physical Chemistry Chemical Physics. 2012;14:2617-30.

[3] Qiu X-Y, Zhuan Q-C, Zhang Q-Q, Cao R, Qiang Y-H, Ying P-Z, et al. Investigation of layered LiNi1/3Co1/3Mn1/3O2 cathode of lithium ion battery by electrochemical impedance spectroscopy (vol 687, pg 35, 2012). Journal of Electroanalytical Chemistry. 2013;688:392-.

[4] Dong H, Yu H, Wang X, Zhou Q, Feng J. A novel structure of scalable air-cathode without Nafion and Pt by rolling activated carbon and PTFE as catalyst layer in microbial fuel cells. Water Research. 2012;46:5777-87.